



# Evaluation of atmospheric nitrogen inputs into marine ecosystems of the North Sea and Baltic Sea – part A: validation and time scales of nutrient accumulation

Daniel Neumann[1], Matthias Karl[2], Hagen Radtke[1], and Thomas Neumann[1]

[1]Leibniz-Institute for Baltic Sea Research Warnemünde, Seestr. 15, 18119 Rostock, Germany
[2]Institute of Coastal Research, Helmholtz-Zentrum Geesthacht, Max-Planck-Str. 1, 21502 Geesthacht, Germany

**Correspondence:** Daniel Neumann (daniel.neumann@io-warnemuende.de)

**Abstract.** The North Sea and the Baltic Sea are impacted by several anthropogenic activities, which put pressure onto the marine ecosystem. One of these pressures is the input of nitrogen compounds, which act as nutrients for phytoplankton growth and induce eutrophication. Atmospheric deposition is a relevant contributor to the marine nitrogen budget, making up $20\%$ to $40\%$ of the nitrogen input of the North Sea and Baltic Sea. But the concentrations of dissolved and particulate nitrogen in the sea are not only determined by the input, but also by the residence time of nitrogen in the system before it is removed by biogeochemical processes or physical advection. Our study aims to estimate the contribution of atmospherically deposited nitrogen to the nitrogen pools of North Sea and Baltic Sea. The contribution of atmospheric nitrogen deposition to dissolved inorganic nitrogen and to particulate organic nitrogen in the surface water was evaluated for both Seas in this study showing the relevance of deposition. Both seas differ significantly with respect to the residence time of water and nutrients. Hence, both water bodies were compared with respect to the accumulation of atmospheric nitrogen. Model simulations with the coupled physical biogeochemical model HBM-ERGOM were performed for this purpose. The fate of atmospheric nitrogen deposition was traced in the marine ecosystem. The model-predicted relevant nutrient concentrations in the surface layer compared well to measurements. Nutrient and oxygen concentrations in deep parts of the Baltic Sea were not properly reproduced but did not impact the simulation quality of surface layer concentrations. The denitrification in the Wadden Sea was underestimated by the model. Tagged dissolved inorganic nitrogen (DIN) with nitrogen from atmospheric deposition reaches a steady-state in the southern North Sea after two years of simulation. This is consistent with the published residence time of nutrients in this region. In contrast, in the Baltic Sea region, the atmospheric nitrogen shares increased year-by-year reaching a steady-state not before the fifth year. This is also consistent with published studies on the residence time of riverine nitrogen in the Baltic Sea. Atmospheric nitrogen shares were evaluated in detail in the second part of this study.

## 1 Introduction

The North Sea and the Baltic Sea are both regional seas, which are heavily impacted by diverse anthropogenic activities (Andersen et al., 2013; Huiskes and Rozema, 1988; OSPAR, 2010a; Andersen et al., 2015; Korpinen et al., 2012; Svendsen et al., 2015). Most impacts have negative consequences for the marine ecosystem and, on the long run, on humans themselves. The



excessive input of nutrients into the seawater, which leads to eutrophication and oxygen depletion (OSPAR, 2017a; Svendsen et al., 2015; Theobald et al., 2009), is one of these negative impacts. The nutrients of highest relevance for this region are bioavailable nitrogen and phosphorus compounds. They are introduced via riverine input, atmospheric deposition, and, to a lesser extent, by oceangoing vessels (OSPAR, 2017b, c; Svendsen et al., 2015; HELCOM, 2015). Particularly riverine nutrient

loads were targeted by several EU Directives in the 1990s (EU-91/271/EEC, 1991; EU-91/676/EEC, 1991; EU-2000/60/EC, 2000). Through reduced nutrient loads of rivers the eutrophication status has been considerably improved in the past three decades (Andersen et al., 2017; Svendsen et al., 2015; Gustafsson et al., 2012). Atmospheric nitrogen deposition into the North Sea and the Baltic Sea has been reduced by roughly $30\%$ in the past two decades (OSPAR, 2017a; Svendsen et al., 2015). However, a Good Environmental Status (GES) is not restored yet (e.g., HELCOM, 2009; Ferreira et al., 2010; OSPAR,

2009, 2017a). Hence, eutrophication is still in the focus of international regional activities – i.e. Helcom's Baltic Sea Action Plan (BSAP), OSPAR's Eutrophication Strategy within the North-East Atlantic Environment Strategy (NEAE Strategy), and descriptor 5 of the Marine Strategy Framework Directive (MSFD) (EU-2008/56/EC, 2008; HELCOM, 2007; OSPAR, 2010b).

Although rivers contribute a large share of external nutrients, the atmospheric nitrogen deposition is not a negligible contributor to the marine nutrient load. In the Baltic Sea region, approximately $20\%$ to $35\%$ of the bioavailable nitrogen input

originates from atmospheric deposition (HELCOM, 2013a, b; Stipa et al., 2007). In the North Sea region, it is approximately $25\%$ to $40\%$ (OSPAR, 2017a, c, d). Other studies found an atmospheric nitrogen contribution of $6\%$ to $8\%$ for the southern North Sea only (Rendell et al., 1993; Troost et al., 2013). Important contributors to atmospheric emissions in coastal and marine regions and, hence, to nitrogen deposition over the ocean are the shipping sector (Geels et al., 2012; Matthias et al., 2010; Aulinger et al., 2016; Aksoyoglu et al., 2016; Tsyro and Berge, 1998; Jonson et al., 2015) and the agricultural sector –

including animal husbandry (Hendriks et al., 2016; Backes et al., 2016; Skjøth et al., 2004; Skjøth et al., 2011; Theobald et al., 2009; Zhang et al., 2008).

The nitrogen deposition might be further reduced in the next decades because emissions of nitrogen oxides ($NO_X$) and ammonia in Europe are in the focus of two international treaties and further individual measures. Namely, these are the Gothenburg Protocol (UNECE, 1999), the EU Directive on *reduction of national emissions of certain atmospheric pollutants*

(EU-2016/2284, 2016), and the Nitrogen oxide Emission Control Areas (NECAs) for the shipping sector. The $NO_X$ emissions of the EU-28 member states are planned to be reduced by $42\%$ to $63\%$ until 2020 with respect to 2005. The ammonia emissions are planned to be reduced by $6\%$ to $19\%$ in the North Sea region (OSPAR, 2017a). IIASA (2011) estimated even higher reductions. Geels et al. (2012) also expected considerable nitrogen emission reductions of up to $16\%$ for Europe until 2020.

The contribution of riverine nutrient input to the marine nutrient budget and its impact on the marine ecosystem dynamics

has been evaluated in the past (Radtke et al., 2012; Dulière et al., 2017; Ménesguen et al., 2007; Wijsman et al., 2004; Blauw et al., 2006a; Los et al., 2014; Ménesguen et al., 2018). However, most studies on source apportionment of atmospheric nitrogen deposition into North Sea and Baltic commonly only consider the deposited amounts of nutrients (e.g., Bartnicki et al., 2011; Tsyro and Berge, 1998; Hongisto, 2014; Theobald et al., 2009; HELCOM, 2009; Aksoyoglu et al., 2016; Stipa et al., 2007). But, their fate and processing in the marine environment are rarely considered (e.g., Dulière et al., 2017; Shou et al., 2018;

Raudsepp et al., 2013; Troost et al., 2013; Ménesguen et al., 2018).




Moreover the spatial and the temporal input patterns of rivers and atmospheric deposition differ considerably from each other. Riverine inputs take place only at specific locations along the coast line and are strongest in spring after the thawing period. In contrast, atmospheric deposition occurs everywhere but is strongest in the vicinity of the coast. Particularly oxidized nitrogen deposition is highest in summer when nutrients are depleted in the ocean (e.g., Stipa et al., 2007), which increases the
phytoplankton growth (Troost et al., 2013).

Based on this state of knowledge we derived two questions which we intended to answer by a modeling study with the coupled HBM-ERGOM model system (Maar et al., 2011; Brüning et al., 2014; Neumann, 2000; Neumann et al., 2002).

a) Is atmospheric nitrogen deposition actually a relevant contributor to marine DIN and biomass?

b) What time scales need to be considered when the contribution of atmospheric nitrogen deposition to the marine nitrogen
budget is to be evaluated?

First, the share of nitrogen deposition in the total nitrogen input is not necessarily equal to the share of deposited nitrogen in nitrogen bound as marine biomass. Additionally, the spatio-temporal nutrient release patterns of riverine inflows and atmospheric deposition differ. Therefore, the contribution of atmospheric nitrogen deposition to marine ecosystem parameters, i.e. dissolved inorganic nitrogen (DIN) and chlorophyll-a, is to be evaluated. For this purpose we used atmospheric nitrogen
deposition calculated by the chemistry transport model (CTM) CMAQ (Karl et al., in prep., a; Appel et al., 2017) as input data for ecosystem model simulations with the coupled HBM-ERGOM model system. This publication presents the nitrogen deposition data, the validation of the model system, and a brief evaluation of question (a). An accompanying publication discusses question (a) in more detail (Neumann et al., 2018b).

Furthermore, the North Sea and Baltic Sea are quite different with respect to nutrient limitations and life time of nutrients
(Lenhart and Pohlmann, 1997; Beddig et al., 1997; Pätsch et al., 2010; Radtke et al., 2012; Reid et al., 1990). It might take several years until a steady-state of atmospheric nitrogen in the seawater is reached (Los et al., 2014). A reasonable time span for a spin-up of marine biogeochemical simulations with tagged nitrogen from atmospheric sources should be estimated.

## 2   Materials and Methods

### 2.1   Atmosphere

The atmospheric emissions, the chemistry transport model simulations, and the deposition data will be presented and validated in detail in Karl et al. (in prep., a). Therefore, they are only briefly described below.

#### 2.1.1   Atmospheric Emissions

Land-based emission data were created by SMOKE for Europe (Sparse Matrix Operator Kernel Emissions; Bieser et al., 2011) on hourly basis on a $5 \times 5 \, \mathrm{km}^2$ grid covering Europe, the Northwest Atlantic Ocean, and the northern coast of Africa. SMOKE
for Europe uses emission data of different sources – primarily from the EMEP Centre on Emission Inventories and Projections





(CEIP). EMEP abbreviates the European Measurement and Evaluation Programme. The emissions are spatially distributed based on demographic, road (Open Streetmap), and land-use data. Point sources are considered and a plume rise model is included. The temporal distribution is performed on the basis of known diurnal, weekly, and annual emission profiles. The emissions were mapped onto the model grid resolutions of $64 \times 64 \, \mathrm{km}^2$ and $16 \times 16 \, \mathrm{km}^2$ (Karl et al., in prep., a).

Shipping emissions were calculated according to Jalkanen et al. (2012). Both methods are based on data of the automatic identification system (AIS). Ocean-going vessels of a Gross tonnage (GT) above 300 and all ocean-going passenger vessels have to be equipped with AIS transceivers. This is required by the International Maritime Organization (IMO). These AIS transceivers send information, such as unique identification number, position, course, and speed, via radio transmission to surrounding AIS receivers. Based on AIS data – position and speed – and a vessel database, which includes age, engine type,
and other vessel specific details, the ship emissions are estimated.

Sea salt emissions were calculated online by the parameterization of Gong (2003) (Kelly et al., 2010). Sea salt surf zone emissions were deactivated because of considerable overestimations in some coastal regions (Neumann et al., 2016b). Natural marine emissions other than sea salt were not considered.

### 2.1.2    Atmospheric Modeling

The meteorological forcing data for the air quality modeling were taken from the coastDat3 atmosphere data set (HZG, 2017). The meteorological simulations were performed with COSMO-CLM version 5.00_clm8 with spectral nudging (Rockel et al., 2008) on a rotated grid of spatial resolution of 0.11 degree (rotated North Pole located at 162° W, 39.25° N). The coast-Dat3 data set is a regional reanalysis for Europe. The meteorological data were processed by a modified version of CMAQ's Meteorology-Chemistry Interface Processor (MCIP) (Otte and Pleim, 2010) to serve as input data for the CTM simulations.

The atmospheric chemistry simulations were performed with the Community Multiscale Air-Quality (CMAQ) model v5.0.1 (Nolte et al., 2015; Foley et al., 2010; Appel et al., 2017). CMAQ calculates atmospheric concentration (gas phase and particle phase), wet deposition, and dry deposition of air pollutants. Figure 1 provides an overview over the most relevant processes. The cb05tump mechanism (Carbon Bond V with toluene and chlorine chemistry) was employed for the gas phase chemistry (Sarwar et al., 2007; Whitten et al., 2010; Yarwood et al., 2005) and aero5 for the aerosol chemistry. The aero5 is based on the
ISORROPIA v1.7 mechanism (Fountoukis and Nenes, 2007; Sarwar et al., 2011) and considers the condensation of $H_2SO_4$, $HNO_3$, $HCl$, and $NH_3$ onto particles and the re-volatilization of the latter three substances. The atmospheric particles are represented by three particle size distributions denoted as size modes (Binkowski and Roselle, 2003): Aitken, accumulation, and coarse modes. These modes represent particles of the size classes from 0.01 to 0.1 μm, from 0.1 to 2 μm, and from 2 to 20 μm in diameter, respectively. Aitken mode particles form from coagulation of gas phase species (nucleation). They grow
by further condensation of gas phase species or by coagulation with other atmospheric particles. Accumulation mode particles arise from growing Aitken mode particles or are directly emitted. Coarse mode particles are considered to be only emitted. Examples for coarse particles are sea salt, brake abrasion, and dust particles.

The dry deposition is calculated according to an updated version of Binkowski and Shankar (1995) and Binkowski and Roselle (2003). The parameterization considers gravitational settling, aerodynamic resistance above the canopy, and surface



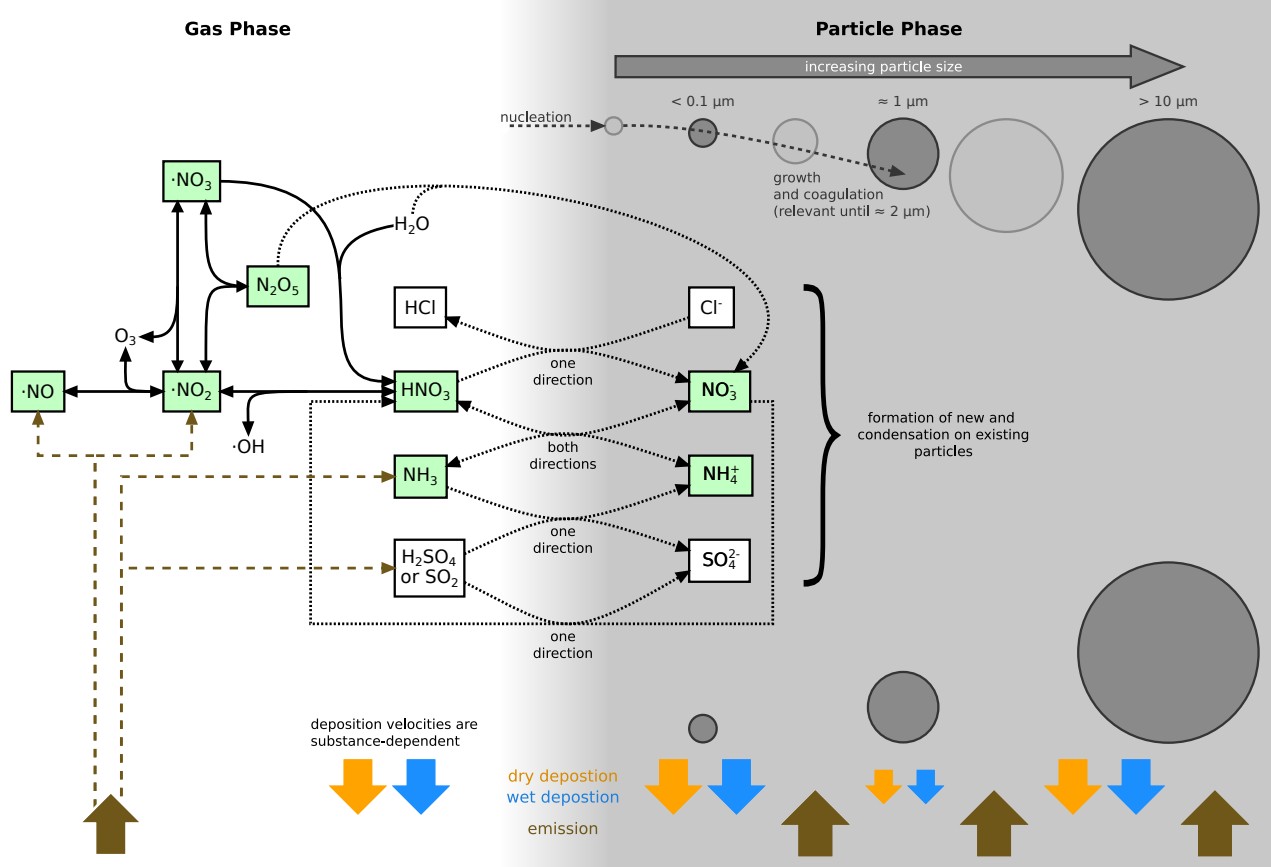

**Figure 1.** Dominant processes and state variables related to the atmospheric nitrogen cycle in the gas phase and particle phase in CMAQ (Community Multiscale Air Quality). Nitrogen-containing state variables with minor relevance are not shown. The three dark grey circles indicate the three particle size modes in CMAQ: Aitken, accumulation, and coarse mode. The light grey circles in between should indicate that in reality a continuous particle size spectrum exists and that the mean model diameters can vary. Most processes related to compounds in the particle phase may occur at particles of all sizes classes. The wet particle phase is not shown here. Nitrogen compounds are colored in light green. Black solid arrows indicate chemical reactions and black dotted arrows indicate phase shifts. Emissions, dry deposition, and wet deposition are colored in brown, yellow, and blue, respectively. The different sizes of the deposition arrows in the particle phase indicate different deposition velocities.

resistance. Aitken mode particles move randomly (Brownian motion), have a high probability to hit another object and, hence, have a high dry deposition velocity. Coarse particles are very heavy, are strongly affected by gravitational setting, and, hence, have a high dry deposition velocity. Accumulation mode particles are less affected by both processes, have the lowest dry deposition velocity, and the longest atmospheric residence time (Seinfeld and Pandis, 2016). The dry deposition velocity of

5    gas phase species depends on their stickiness to surfaces (higher stickiness = higher dry deposition velocities) – i.e. ammonia ($NH_3$) has a high stickiness. The wet deposition parameterization is described in Foley et al. (2010). Accumulation mode

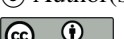



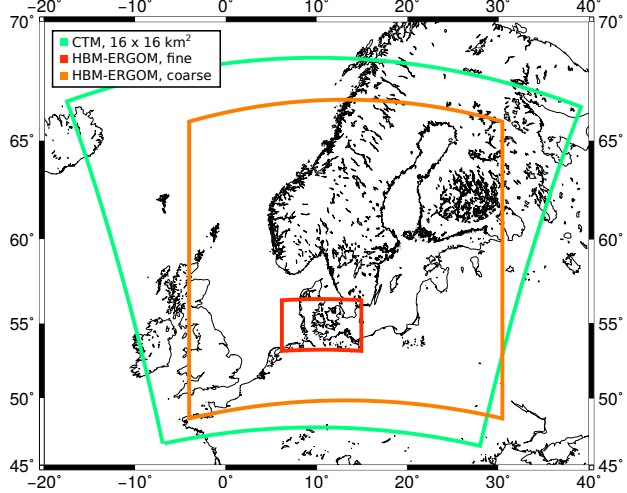

**Figure 2.** Model domains. The extent of the inner model domain of the chemistry transport model (CTM) is plotted in green. The extents of the HBM-ERGOM model domains of $5' \times 3'$ (coarse) and $50'' \times 30''$ grid cell size (fine) are represented by the orange and red frames, respectively.

particles are also expected to have the lowest wet deposition velocity of the three sizes modes. The wet deposition velocity of gas phase species also is substance specific.

A simulation was performed for the year 2012 on a grid of $16 \times 16 \ \mathrm{km}^2$ resolution, which covers the North Sea, the Baltic Sea and the adjacent land masses (Fig. 2). It was one-way nested into a grid of $64 \times 64 \ \mathrm{km}^2$ resolution covering Europe, 5 the Northwest Atlantic Ocean and northern African coast. Both grid domains have 30 vertical z-layers. The lateral boundary conditions were taken from FMI APTA global reanalysis (Sofiev et al., 2018). The spin-up period was 30 days, which is sufficient for a regional chemistry transport model without online meteorology. Output data were written on an hourly interval.

### 2.1.3 Deposition

All oxidized and reduced nitrogen species of the hourly wet and dry deposition CMAQ output were summed. These are the 10 following variables (not all are shown in Fig. 1):

– Oxidized nitrogen: NO, $NO_2$, $HNO_3$, $N_2O_5$, $NO_3^-$, $NO_3$, PAN (peroxyacetyl nitrate), HONO, PNA (peroxynitric acid; only wet deposition)

– Reduced nitrogen: $NH_3$, $NH_4^+$

Daily mean deposition fields of reduced and oxidized nitrogen were calculated and bilinearly interpolated onto the HBM-15 ERGOM model grids as model input.

The default atmospheric phosphate deposition of HBM-ERGOM was used, which is constant: $0.471 \ \mu\mathrm{mol} \ \mathrm{m}^{-2} \ \mathrm{d}^{-1}$ ($\approx$ $5.33 \ \mathrm{mg} \ \mathrm{m}^{-2} \ \mathrm{a}^{-1}$).

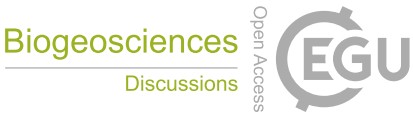

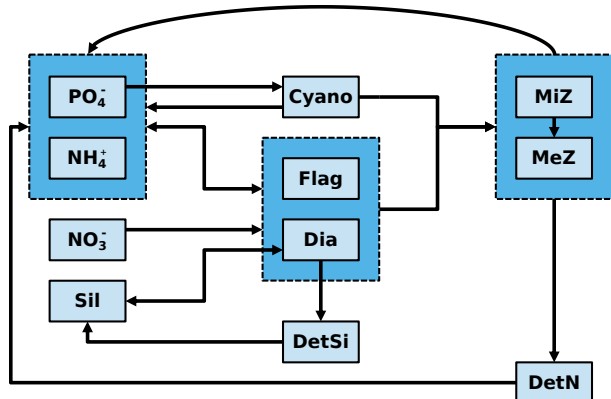

**Figure 3.** ERGOM water column state variables except for oxygen. Cyanobacteria take up $N_2$ from atmospheric mixing, which is not shown. The interaction with the sediment is also not shown. A detailed model description including all parameters is given in the supplement. Figure taken from Neumann et al. (2018a).

## 2.2 Ocean

### 2.2.1 Oceanic Modeling

The ocean physics were modeled by the HIROMB-BOOS-Model (HBM) (Brüning et al., 2014; Poulsen et al., 2015), which is based on the BSH circulation model (BSHcmod) originally developed by the Federal Maritime and Hydrographic Agency (BSH) of Germany (Dick and Kleine, 2007). The HBM is used by German, Danish, Swedish, and Finish governmental agencies for operational ocean current and sea level prediction as well as for storm surge warning services. In addition, it will be used as model for the Copernicus Marine Environment Monitoring Service (CMEMS) forecast product until 2019. A validation of the physical model parameters is not presented in this publication to reduce the length of this article. HBM has been extensively validated for CMEMS. Validations were published by Wan et al. (2012) and Brüning et al. (2014). Wan et al. (2012) found drifting salinity at the sea floor in deep regions of the Baltic Sea ($> 75\,\mathrm{m}$ depth) in a two-year free run, which might indicate a too strong vertical mixing. Sea surface temperature and salinity are generally well predicted.

### 2.2.2 Biogeochemical Modeling

For the biogeochemical studies, the Ecological ReGional Ocean Model (ERGOM) is coupled to HBM. ERGOM was developed for modeling biogeochemical processes in the Baltic Sea (Neumann, 2000; Neumann et al., 2002). It has been used in several studies focusing on the Baltic Sea in the past 15 years (e.g., Kuznetsov et al., 2008; Lessin et al., 2014; Miladinova and Stips, 2010; Neumann et al., 2015; Radtke et al., 2012; Neumann and Schernewski, 2005; Schernewski and Neumann, 2005). Different model branches of ERGOM exist.





The coupled HBM-ERGOM model system has been applied in a few studies before (Maar et al., 2011; Wan et al., 2012). However, in this study's version, some parameters were modified and it has been extended by an additional tracer, labile dissolved organic nitrogen, according to Neumann et al. (2015).

The model consists of 13 state variables representing nutrients, plankton, and detritus in the water column and organic matter in the surface sediment. Additional tracers in the water column are oxygen and labile dissolved organic nitrogen. An overview is shown in Fig. 3.

Nutrients are nitrate, ammonium, phosphate, and silicate. Silicate is a limiting nutrient in the North Sea but not in the Baltic Sea. Hence, it is commonly not considered in Baltic Sea studies. During the respiration process in the model, plankton, releases a mixture of ammonium ($90\,\%$) and labile dissolved organic nitrogen (LDON, $10\,\%$). The latter is not bioavailable but is degraded to ammonium. It is needed for calculating light attenuation (Neumann et al., 2015). Phytoplankton species are divided into three functional groups – diatoms, flagellates, and cyanobacteria – and zooplankton into two – micro- and meso-zooplankton. Detritus is divided into normal detritus and silicon detritus. Plankton and normal detritus are represented in nitrogen units. Phosphorus is coupled to nitrogen via a modified Redfield ratio (N:P) of $1 : 0.072$ ($\approx 13.9 : 1$). Silicon is coupled to nitrogen in a ratio of $1 : 0.94$ (N:Si). Sediment tracers are divided into benthic nitrogen and benthic silicon, whereas phosphorus is coupled to nitrogen via the Redfield ratio. Iron reduction and release of phosphate under anoxic conditions in the sediment are not represented in this ERGOM version (Gustafsson and Stigebrandt, 2007; Sundby et al., 1992). The sediment consists of one layer. A further description in provided in the supplement.

### 2.2.3 Source Attribution

A method for tagging elements of specific sources – such as *"nitrogen from rivers"* – was utilized in this study to track atmospheric nitrogen deposition. The method was described by Ménesguen et al. (2006) and implement in ERGOM by Neumann (2007) and Radtke et al. (2012). It allows attributing tracer concentrations to specific predefined sources. The method is also denoted as TBNT (trans-boundary nutrient transport) method.

If the contribution of one source, such as the atmosphere, to the budget of one element, such as nitrogen, is to be considered, all tracers containing the particular element are duplicated. One set of tracers represents the total tracer concentrations, such as (total) *"nitrate"* or (total) *"diatoms expressed as nitrogen"*. The other set represents the tagged element's tracer concentrations, such as *"nitrate with nitrogen from atmospheric deposition"* or *"diatoms expressed as nitrogen from atmospheric deposition"*. This method allows tracking different nutrients released by diverse sources. Moreover, it allows (a) tracking elements that have undergone specific processes, such as nitrogen that was assimilated into plankton at least once, and (b) deriving the residence time of specific elements in the system (see Radtke et al. (2012)). In order to improve the readability of the text, the sources of tagged tracers are written with the source's name as subscript: *"nitrogen from atmospheric deposition in nitrate"* is written as nitrate$_{\text{atmos}}$.





### 2.2.4 Model Setup

The North Sea and Baltic Sea are covered by a model domain of $5' \times 3'$ (lon $\times$ lat) resolution, into which a $50'' \times 30''$ resolved domain is two-way nested (Fig. 2). The latter covers the German territorial waters. The vertical layers are represented by z-star coordinates (spatio-temporally variable layer thickness): the coarse grid has 36 and the fine grid has 25 layers. River data –

runoff and nutrient loads – and boundary conditions to the Atlantic Ocean were taken from the BSH default setup. Current and tidal boundary conditions for open boundaries to the Northeast Atlantic are calculated by a 2D physical model of the Northeast Atlantic Ocean (Brüning et al., 2014). Salinity and temperature boundary condition are climatological data (Janssen et al., 1999; Maar et al., 2011). The meteorological conditions at the sea surface are taken from operational weather forecasts of the German Weather Service (DWD). Initial conditions for HBM were generated from a regular model run of the BSH. No further

spin-up period for the model physics was necessary. The biogeochemical boundary conditions for the Northeast Atlantic Ocean are climatological data based on World Ocean Atlas (WOA05) as described by Maar et al. (2011) except for LDON. LDON boundary condition data depend on the ammonium boundary conditions. Initial conditions for ERGOM were generated from a two-month spin-up period in November and December 2011. These initial conditions did not contain tagged tracers.

The convergence of tagged tracer concentrations towards a steady-state takes some years (e.g., Los et al., 2014). Therefore,

the biogeochemical model was run for five years with activated tagging. The physical model and the silicate tracer concentrations were restarted from the initial conditions each year (details in next paragraph). The atmospheric and river forcing were also repeated each year. The biogeochemistry does not interact with the physics. Hence, the physical variable fields were the same in each year. This setup is convenient for the evaluation of the propagation of the tagged tracers in the model domain: inter-annual variations are only due to the biogeochemistry and not caused by changes in the model forcing or by the ocean

physics. In the following, we denote the five years as five iterations.

In preliminary simulations without the restart of silicate each year, modeled silicate concentrations in the Baltic Sea decreased significantly over the simulated period, indicating a missing source. In contrast to reality, silicate even became a limiting nutrient for diatom growth in the western part of the Baltic Sea from the fourth simulated year on. Because the computing resources to evaluate and optimize this issue were missing, it was decided to reset silicate each year as workaround.

## 2.3 Presentation of results

The biogeochemistry is validated by comparing model results of iteration 1 with measurements (see Sect. 2.4). Model values are taken from the same depths from which measurement data were available. Time series of surface water concentrations and vertical profiles are plotted for the validation. Evaluated tracers and stations are presented in Sect. 2.4.

The mean model values of DIN and chlorophyll-a in the upper five model layers are considered and plotted for the evaluation

of the atmospheric deposition. The upper five model layers approximately correspond to the top 12 m of the water column. The depth of these layers is not constant because of the vertical z-star coordinates used.




**Table 1.** Databases of measurement data used for model validation. The column *region* indicates regions, for which data were extracted from the databases for model validation. Data of further regions is included in these databases but was not extract.

| Abbreviation | Full Name | Region |
| --- | --- | --- |
| ICES | Data portal of the International Council for the Exploration of the Sea | North Sea |
| DOD | Database of the Deutschen Ozeanographischen Datenzentrum (DOD, German Oceanographic Data Center) | North Sea |
| HELCOM | Data collected by HELCOM member states and provided via the Baltic Sea (HELCOM) Monitoring data portal of ICES | Baltic Sea |
| IOWDB | Oceanographic Database of Leibniz Institute for Baltic Sea Research Warnemünde | Baltic Sea |

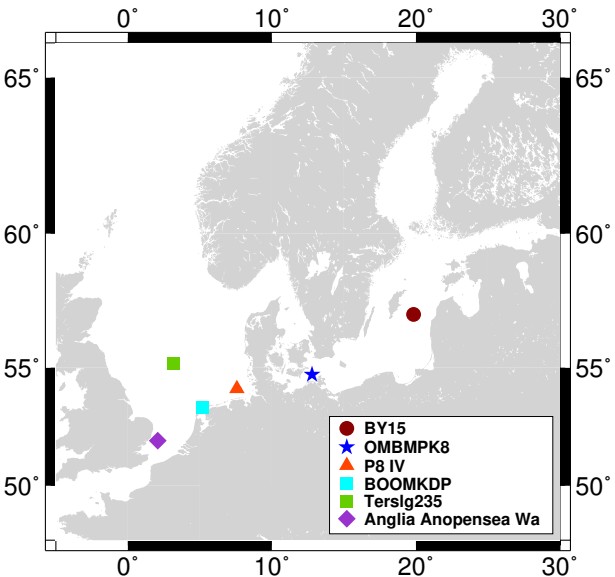

**Figure 4.** Stations for model validation and evaluation. The station Terslg235 is only used for evaluation in part B of this study.

## 2.4 Observational data for model validation

The biogeochemical model output was validated with observational data from four different databases covering the North Sea and Baltic Sea (Table 1). Modeled DIN (dissolved inorganic nitrogen), DIP (dissolved inorganic phosphorus), silicate (only North Sea), oxygen, and chlorophyll-a concentrations were compared with measurements if available. The comparisons at five measurement stations, which are considered to be representative for their respective region, are presented in the validation section (Fig. 4). We did not consider stations in the vicinity of the coast for the validation because inaccuracies of the land-sea mask might induce artifacts in the nitrogen deposition along the coast line (e.g., Neumann et al., 2018a; Hongisto, 2014).

The presented Baltic Sea stations are OMBMPK8 and BY15. OMBMPK8 is located in the western Baltic Sea and in the vicinity of heavily anthropogenically influence land masses. Hence, it is inconsiderably impacted by riverine nutrient loads.



BY15 is located in the eastern Gotland Basin, which is one of the major basins of the Baltic Sea. It is farther distant from anthropogenic activity than OMBMPK8 but still impacted – i.e. by shipping activity. Both stations are part of the regular monitoring activities of the Leibniz Institute for Baltic Sea Research.

The considered North Sea stations are Anglia Anopensea Wa, BOOMKDP, and P8 IV. Anglia Anopensea Wa is located between the United Kingdom and Belgium northeastward of the English Channel. It is close to the British coast and affected by the Thames river plume rather than by the Rhine river plume. BOOMKDP is located in the Dutch waters close to the Dutch coast. Although quite distant from the estuary of the Rhine, it is still impacted by its plume. Finally, P8 IV is located in the central German Bight and outside of the Wadden Sea. It is affected by nutrient loads of the Elbe River. Both stations BOOMKDP and P8 IV are located in the vicinity of the European mainland and, hence, impacted by the deposition short-live atmospheric pollutants. The measurements at Anglia Anopensea Wa, and BOOMKDP were extracted from the ICES measurement database. The measurements at P8 IV were provided by the BSH from their DOD database (Table 1).

The number of data points per tracer was too low for a meaningful statistical evaluation. Instead, a graphical comparison is performed.

## 3    Results and Discussion

The results section is structured in three subsections. The Sect. 3.1 deals with the atmospheric nitrogen deposition and briefly describes it. Section 3.2 deals with the validation of the ERGOM model results. In Sect 3.3 the time scales, until which a steady-state of atmospheric nitrogen is reached, are estimated.

### 3.1    Atmospheric Deposition: Overview

In this section, the atmospheric nitrogen deposition into the North Sea and Baltic Sea is briefly assessed (details in: Karl et al., in prep., a). Figure 5 gives an overview of the total annual average nitrogen deposition and Table 2 provides corresponding numbers divided into wet & dry, reduced & oxidized, and North Sea & Baltic Sea annual average nitrogen deposition.

The nitrogen deposition is highest at the southern coasts of the North Sea and Baltic Sea, along the British east coast, and at the southern tip of Norway. Dry deposition of coarse particles and sticky gaseous substances creates the steep gradients of nitrogen deposition close to the coastline. The deposition is enhanced by interaction between gaseous nitrogen species and coarse sea salt particles (Neumann et al., 2016a). In contrast, the wet deposition is responsible for the patchy patterns in the open sea – the patches being the result of individual precipitation fronts.

Deposition in coastline grid cells might also be increased artificially. Dry deposition over land is higher than over sea. The nitrogen deposition to the ocean is calculated during the post-processing of CMAQ model data via the land-sea-fraction for each grid cell. However, the enhanced deposition over land is not considered in this process. Thus, the sea-fraction of the nitrogen deposition is overestimated and the land-fraction is underestimated. Moreover, the grid resolution of the CMAQ simulations was coarser than that of the HBM-ERGOM simulations adding another source for possible under- or overestimation –

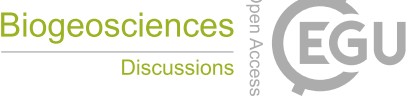



**Table 2.** Dry, wet, and total deposition (each three columns side by side) of oxidized, reduced, and total nitrogen (rows) into the North Sea, Baltic Sea, and both seas (three superior columns) in the year 2012. Data are calculated from the nitrogen deposition fields interpolated onto the HBM-ERGOM model grid resolution. These values slightly differ from the values presented in Karl et al. (in prep., a) due to different interpolation methods and different target grid geometries. The values are rounded to full integers und, hence, the totals diverge by $\pm 1$ from manually calculated sums.

| nitrogen deposition $\left[\text{kt N a}^{-1}\right]$ | North Sea | | | Baltic Sea | | | Both Seas | | |
|---|---|---|---|---|---|---|---|---|---|
| | wet | dry | **total** | wet | dry | **total** | wet | dry | **total** |
| oxidized | 100 | 68 | 169 | 62 | 53 | 116 | 163 | 122 | 284 |
| reduced | 67 | 54 | 121 | 46 | 27 | 73 | 113 | 82 | 194 |
| **total** | 167 | 123 | 290 | 108 | 81 | 189 | 275 | 203 | 479 |

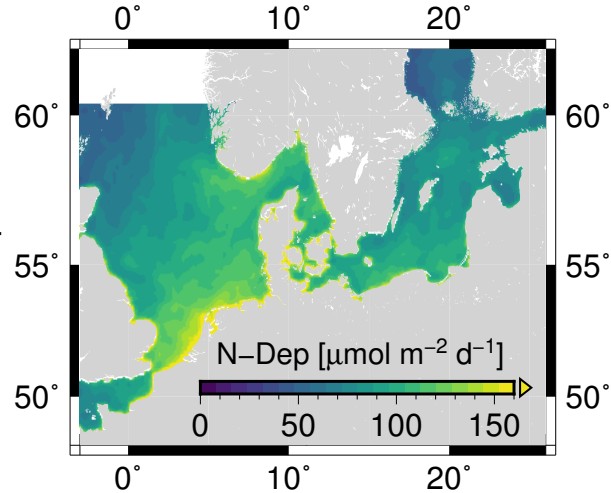

**Figure 5.** Annual average nitrogen deposition in $\mu\text{mol m}^{-2}\text{ d}^{-1}$ into North Sea and Baltic Sea in the year 2012.

depending on the region. Neumann et al. (2018a) evaluates these issues for the western Baltic Sea by comparing two differently resolved nitrogen deposition data sets.

EMEP nitrogen deposition data for the year 2012 are shown in Table 3 for comparison (OSPAR, 2017d; Bartnicki et al., 2017). In the Baltic Sea, the used CMAQ nitrogen deposition is lower than suggested by EMEP data and other studies (Bartnicki and Fagerli, 2008; Langner et al., 2009; Bartnicki et al., 2011). The EMEP total nitrogen deposition is approximately 29 % higher than this study's deposition of the same year, whereby the reduced nitrogen deposition is even 45 % higher.

Karl et al. (in prep., a) performs a validation of the CMAQ simulation results with measurements: modeled atmospheric concentrations of nitrogen compounds do not strongly deviate from measurements. The nitrate wet deposition is underestimated





**Table 3.** EMEP nitrogen deposition of the year 2012 into the North Sea and Baltic Sea as published by OSPAR (OSPAR, 2017d) and HELCOM (Bartnicki et al., 2017), respectively. Absolute deposition is quoted in the columns two and four. Relative numbers with respect to the nitrogen deposition used in this study are stated in columns three and five.

| *nitrogen* | North Sea | | Baltic Sea | |
|---|---|---|---|---|
| *deposition* | absolute | relative | absolute | relative |
| *EMEP* | $[\mathrm{kt\ N\ a^{-1}}]$ | $[\%]$ | $[\mathrm{kt\ N\ a^{-1}}]$ | $[\%]$ |
| oxidized | 263 | 146 | 147.3 | 119 |
| reduced | 210 | 162 | 113.7 | 145 |
| **total** | 473 | 153 | 261.0 | 129 |

by approximately $70\,\%$ in Finland. In south Sweden, the precipitation is underestimated by approximately $25\,\%$ but the modeled nitrate wet deposition is not generally lower than measurements in this region. Hence, there is no evidence for a systematic underestimation of the nitrogen deposition by CMAQ in the central model domain.

Most likely missing nitrate formation or too weak atmospheric nitrate transport are possible reasons for too low nitrogen

deposition in some regions. The differences between CMAQ and EMEP are larger for reduced than for oxidized nitrogen (i.e. nitrate). Thus, differences in the spatio-temporal distribution of ammonia emissions might be a reason because elevated ammonia levels favor nitrate formation. A comparison of ammonia emissions by SMOKE for Europe with reported emissions of the German Federal Environmental Protection Agency (UBA, Umweltbundesamt) indicate underestimations in the ammonia emissions in Northern Germany (Remlinger, 2018). Additionally, the differences between CMAQ and EMEP in the wet depo-

sition may be due to a higher rain frequency in the EMEP simulations (Karl et al., in prep., a). A more quantitative evaluation of all fractions of nitrogen deposition is necessary but not possible given the data currently available.

In the North Sea, the CMAQ nitrogen deposition is considerably lower than predicted by the EMEP model and by other studies (de Leeuw et al., 2003; Hertel et al., 2002; Bartnicki and Fagerli, 2008). It amounts approximately 2/3 of the reported EMEP nitrogen deposition of 2012 (OSPAR, 2017d, Table 2). In another study for the year 2008, CMAQ nitrogen deposition

was also lower than EMEP nitrogen deposition Neumann et al. (2016a). We did not evaluate spatial differences. Validated nitrogen deposition measurements of high spatio-temporal density are not available over seawater for the North Sea. Therefore, a detailed validation of the nitrogen deposition data sets is not possible and it is not clear whether the CMAQ nitrogen deposition is actually too low over sea. The reasons might be: too low emissions in the UK or too low concentrations of particulate nitrogen compounds at the model domain boundaries. The dry deposition velocity or the rate of precipitation might also be too low above

the North Sea.

Because CMAQ and the other deposition data sets are in the same order of magnitude and the spatial pattern is reasonable, we assume that the given data set is sufficiently valid for the proposed usage – keeping in mind that our nitrogen deposition is at the lower limit of previous studies.





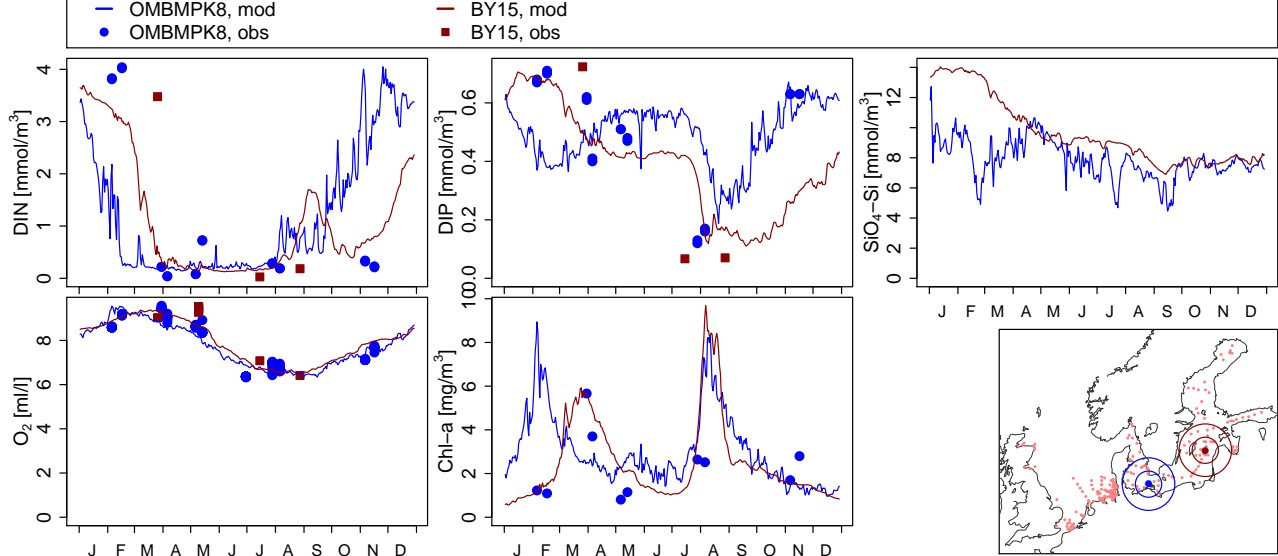

**Figure 6.** Time series of measurement (symbols) and model data (lines; first model iteration) at the stations OMBMPK8 (blue; $54.72°$ N, $12.78°$ E) and BY15 (red; $57.33°$ N, $20.05°$ E). The concentrations of DIN (dissolved inorganic nitrogen), DIP (dissolved inorganic phosphorus), silicate, oxygen, and chlorophyll-a are shown from top left

## 3.2 Validation of the modeled marine biogeochemistry

Model and measurement data at five locations in the North Sea and Baltic Sea are considered for the model validation. Surface concentrations of relevant tracers – DIN, DIP, silicate, $O_2$ and Chlorophyll-a – are compared at all stations. At three stations, vertical profiles are evaluated. Model data of the first iteration are considered. The Baltic Sea data is compared first.

### 3.2.1 Baltic Sea

Figure 6 shows model (lines) and measurement data (symbols) in the western Baltic Sea (OMBMPK8) and the western Gotland basin (BY15).

At the station OMBMPK8, the $O_2$ surface concentration is well represented by the model (Fig. 6, bottom left). The vertical profile of the water column shows that an oxygen minimum in later summer is not reproduced (Fig. 7). A decline of the DIN (dissolved inorganic nitrogen) concentration in spring and its depletion in summer are well represented. However, the model predicts increasing concentrations from August to October reaching peak values of nearly $4\ \mathrm{mmol\ m^{-3}}$. In contrast, the DIN measurements remain constantly below $0.5\ \mathrm{mmol\ m^{-3}}$. Stations in the vicinity to OMBMPK8 show the same discrepancy. The model also over-predicts DIN concentrations below $10\ \mathrm{m}$ depth in this time period. The reason for this is unclear because DIP and Chl-a are well predicted in autumn (see below).





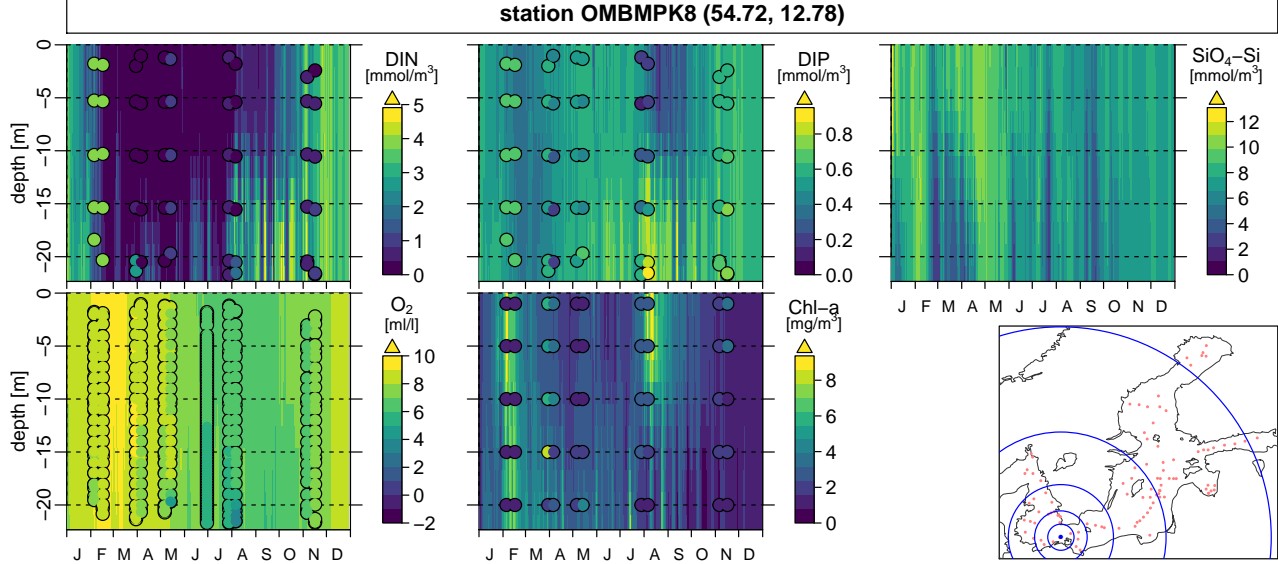

**Figure 7.** Vertical profiles of the same tracers as shown in Fig. 6 at station OMBMPK8. Filled circles indicate measurements. The remaining colored area represents model data.

The modeled DIP concentrations agree well with the measurements. They show a maximum during spring and a minimum in August and September, which is also consistent to the measurements. The vertical profile correctly shows a layer of DIP-rich water above the ground in summer. The chlorophyll-a surface concentrations at OMBMPK8 agree in the magnitude of both the spring bloom peak and the summer minimum. The modeled peak in autumn is not present in the measurements. The temporal

occurrence of the peak concentrations deviates from the measurements. The spring peak arises approximately 1 to 2 months too early in the model. This is also reflected in the vertical profiles. The summer/autumn peaks might also be a few weeks too early. Consequently, the algae blooms start too early in the considered region. Silicate measurement data were not available at this station.

At BY15, the surface oxygen concentration and its annual cycle are matched closely by the model. The modeled winter and

summer concentrations of DIN approximately agree with the measurements. They decrease too early in the spring compared to measurements. It is probably caused by a too early onset of the diatom bloom in spring (first Chl-a peak in spring), although this conclusion cannot be confirmed because chlorophyll-a measurements are missing at this station. A DIN peak in August and September is caused by cyanobacteria (not plotted) but, in reality, the bloom either was weaker or occurred at a later point of time.

The oxygen concentrations are properly represented by the model at most depths and time steps. However, the model fails to predict the absence of oxygen at the sea floor: oxygen concentrations are overestimated. Wan et al. (2012) found underestimations in the bottom salinity at stations close to Gotland (Wan et al., 2012, Fig. 5). They might be caused by too strong vertical mixing leading to increased oxygen concentrations at the sea floor as described above.



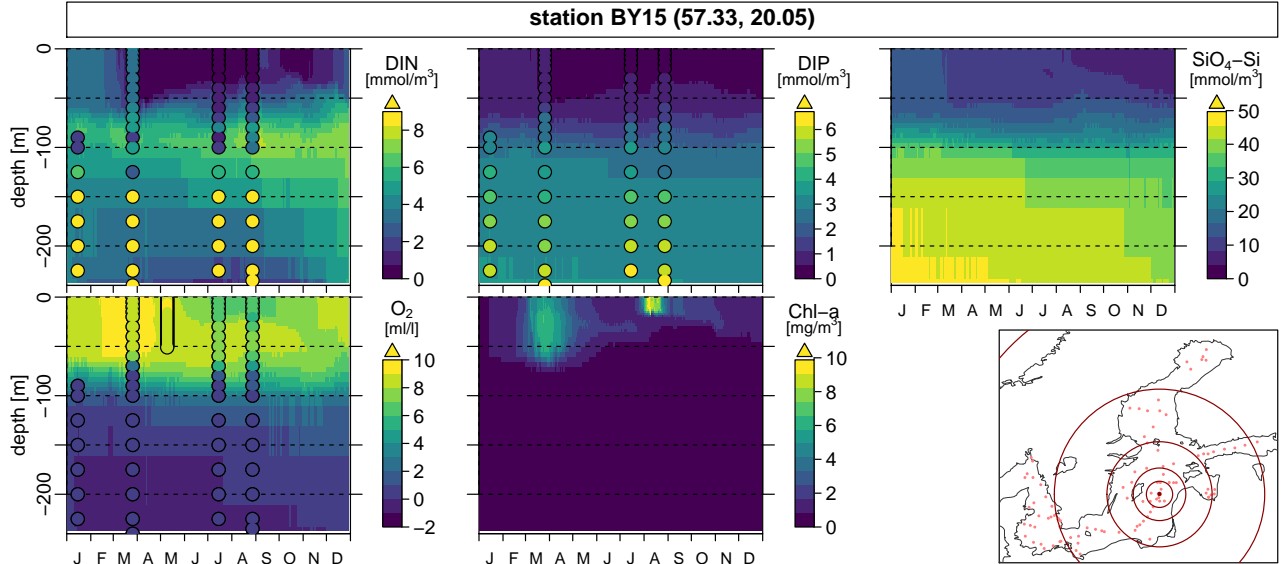

**Figure 8.** Similar to Fig. 7 but showing data at station BY15.

The vertical distribution of DIN concentrations is reproduced by the model until approximately $120\,\mathrm{m}$ depth. A layer of high DIN concentrations was measured at $\approx 80\,\mathrm{m}$ depth in summer and was reproduced at $\approx 100\,\mathrm{m}$ depth by the model. Below that depth, the model predicts decreasing DIN concentrations but considerably increasing DIN concentrations were measured. Looking into individual nitrate ($\mathrm{NO_3^-}$) and ammonium ($\mathrm{NH_4^+}$) data (see Fig. S.1 in the supplement) reveals that the high DIN

concentrations below $120\,\mathrm{m}$ are caused by ammonium: the ammonium concentrations partly exceed $20\,\mathrm{mmol\,m^{-3}}$. The nitrate concentrations, in contrast, were well reproduced by the model in all depth. This issue indicates that the spin-up period was too short for the deep central Baltic Sea basins to converge to realistic values. This issue does not seem to affect the surface layer concentrations, which are in the focus of this study's evaluation.

The modeled and measured DIP concentrations compare well with respect to their annual cycle and their magnitude (Fig. 6).

DIP concentrations are underestimated below $80\,\mathrm{m}$ (Fig. 7). This is related to the missing oxygen minimum at the sea floor and to missing processes for phosphate release in oxygen minimum regions (Gustafsson and Stigebrandt, 2007; Sundby et al., 1992). Chlorophyll-a and silicate concentration measurements were not available.

### 3.2.2 North Sea

Three stations in the North Sea were considered. They are located in the German Bight (P8 IV), at the Dutch coast (BOOMKDP),

and close to the English Channel (Angelia Anopensea Wa).

Measurements of $\mathrm{O_2}$ concentrations are missing at the station P8 IV. The annual cycle and the magnitude of modeled $\mathrm{O_2}$ concentrations are realistic. The model predicts relatively constant silicate concentrations throughout the year and a weak annual cycle. It considerably underestimates the measured silicate concentrations in winter and early spring, which points to



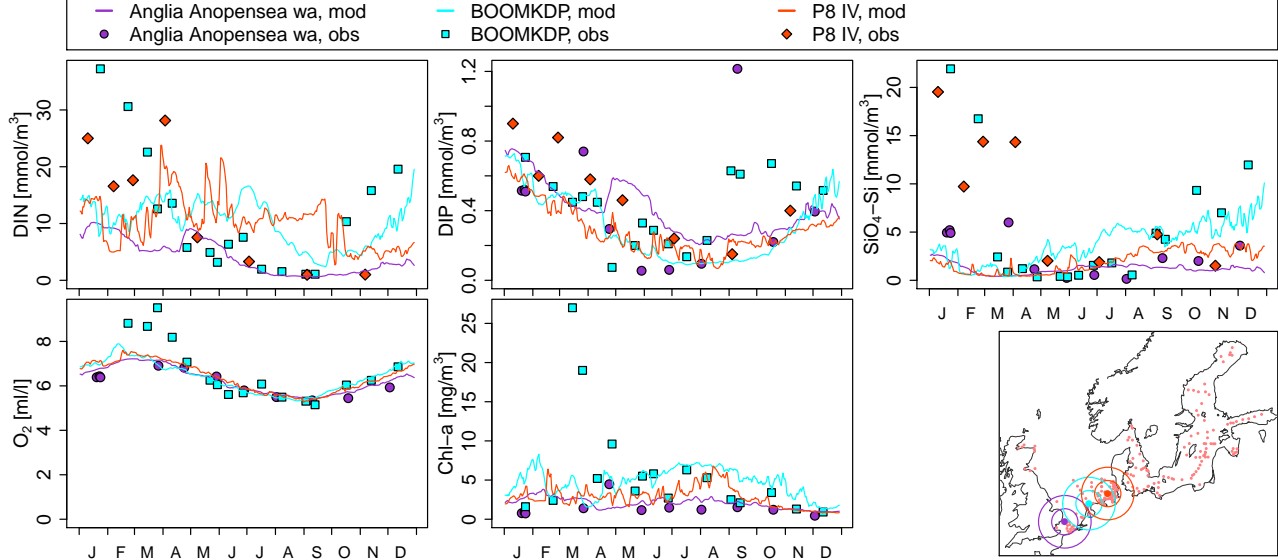

**Figure 9.** Like Fig. 6 but showing data at the stations P8 IV (cyan; 54.15° N, 7.58° E), BOOMKDP (yellow; 53.38° N, 5.17° E), and Anglia Anopensea wa (violet; 51.98° N, 2.07° E).

issues during the spin-up phase. This underestimation probably considerably limits the growth of diatoms in spring. Modeled summer and autumn concentrations agree well with measurements. At 30 m depth, the silicate concentrations show the same pattern (Fig. 10).

The model reproduces the DIN concentrations well in the beginning and at the end of the year (Fig. 9). However, it does
not predict the decline of DIN in spring and its depletion in summer. The DIN concentrations at 30 m depth show the same shortcoming but modeled values are closer to the measurements than at the surface in spring. DIN depletion is lacking in the model in large parts of the southern North Sea. This issue might be caused by underestimating the denitrification in the relatively shallow Wadden Sea. Denitrification is an important DIN removal processes in the Wadden Sea (van Beusekom et al., 1999), which is not properly represented by the simple sediment model. This is also reflected by the absence of a
vertical gradient in the modeled DIN concentrations during summer and autumn. Additionally, a too weak diatom bloom in spring resulting from under-predicted silicate concentration might induce overestimations in DIN. The DIP concentration was slightly underestimated in the beginning of the year. Thus, the spin-up yields to low DIP concentrations. Nevertheless, the concentrations in summer and autumn as well as their annual cycle are well captured by the model. At 30 m depth the model fails to predict a DIP peak in May but, apart from that, reproduces the measurements well.

The station of BOOMKDP is located westward of P8 IV and northward of the Dutch coast. The oxygen concentrations in late winter and early spring are underestimated. However, overall, the model reproduces the oxygen concentration and its annual cycle well. The model nicely predicts increasing DIN and silicate concentrations in autumn. Equally to P8 IV, the DIN depletion in summer is not predicted. Additionally, the DIN and silicate initial concentrations in the beginning of the year are





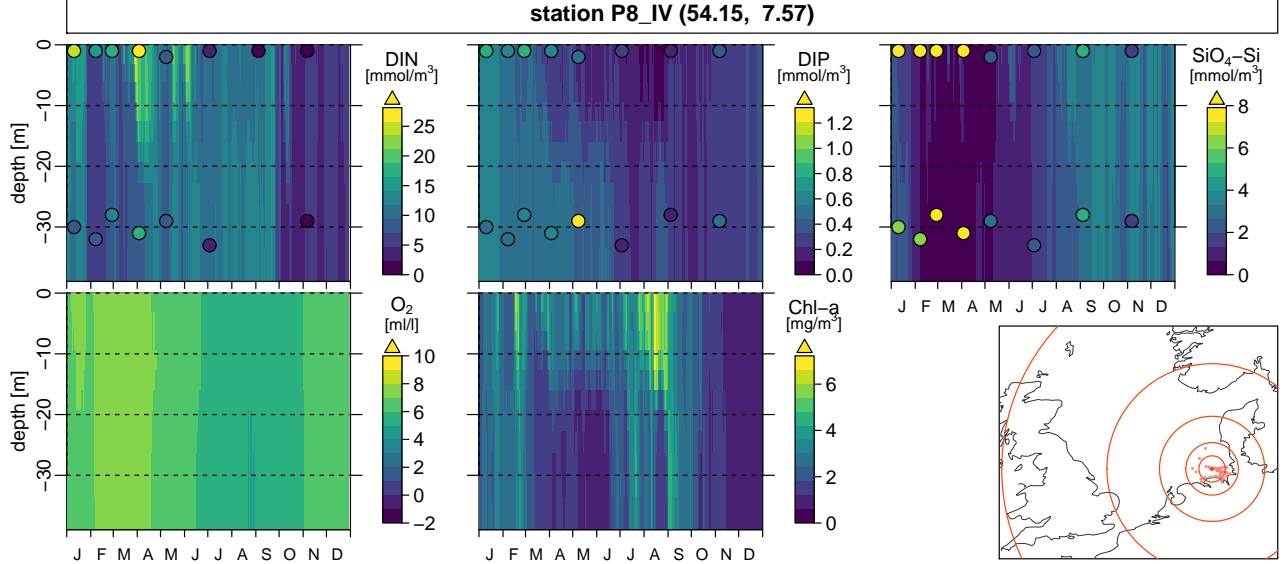

**Figure 10.** Similar to Fig. 7 but showing data at station P8 IV.

too low after the spin-up phase. A comparison of modeled and measured chlorophyll-a concentrations clearly shows that the model does not reproduce the algae bloom in March. This is probably caused by missing silicate as noted for the station P8 IV. As a result, too little DIN is consumed by algae growth. Moreover, missing denitrification in the Wadden Sea might yield too high DIN concentrations similar to P8 IV. The DIP concentrations are well predicted except for autumn: they decrease in
5   spring and increase in autumn but the observation indicate the increase to occur two months earlier than the model does.

The station Angelia Anopensea Wa is located at the English coast eastward of the county of Suffolk. The $O_2$ concentrations and their annual cycle are well represented. Measurements of DIN are not available. The modeled silicate concentrations are fairly constant throughout the year with a slight decrease in February. Model and measurement silicate data are in a similar size range from the end of spring until the end of the year. However, the measurements indicate a slight increase in autumn whereas the modeled values remain on the same level. In winter and early spring, however, the measured silicate concentrations
10   considerably exceed the modeled concentrations and steeply decline from March to April. The chlorophyll-a measurements indicate an algae bloom in April that is missing in the model data. The model, instead, yields algae blooms in February (diatoms) and from July until September (flagellates). The model predicts fairly constant DIP concentrations until June and a decline afterwards. The measurements indicated decreasing concentrations in April and May correlating with the peak in chlorophyll-a. Thus, the modeled DIP concentrations have a time lag compared to observations.

### 3.2.3   Validation Summary

In summary, the model predicts the considered parameters in the correct order of magnitude. The annual cycles of oxygen and DIP are correctly reproduced at the sea surface. Oxygen concentrations at the sea floor are overestimated in the central Baltic





Sea and the model fails to predict oxygen minimum zones. Moreover, the DIN concentrations are underestimated in deeper layers ($> 120\,\mathrm{m}$) of the Gotland basin due to a too short spin-up period. The failed prediction of oxygen minimum zones and of high DIN concentrations below $120\,\mathrm{m}$ is not critical for the model results because the time period of the simulation is too short for a feedback from the bottom water to the sea surface.

DIN concentrations and algae blooms are realistic in the surface layer of the Baltic Sea but the occurrence of algae blooms is partly shifted in time compared to measurements. In contrast in the North Sea, the model fails to reproduce an algae bloom in spring in its full magnitude. This is due to too low silicate concentrations in January after the spin up period. Diatoms, which usually are the first species to bloom each year, need silicate as nutrient (Reid et al., 1990). Either the winter silicate concentrations are generally underestimated by the model or the silicate concentrations are particularly high in winter 2011/2012 but

not captured by the model due to missing processes or sources. A further assessment of this shortcoming is out of the scope of this study. The DIN concentrations in the German Bight are not depleted in summer. First, this is a result of the underestimated phytoplankton bloom, which leads to too less DIN removal in summer. Additionally, missing denitrification is expected to be responsible for the missing DIN depletion in the Wadden Sea, which affects the whole German Bight. Other models also do not fully capture the denitrification in the Wadden Sea – but to a less strong extend as in this study (e.g., Große et al., 2017).

Most biogeochemical models of the North Sea and of the Baltic Sea will perform better for their respective target regions. However, there are only a few models available to model biogeochemical processes in the North Sea and Baltic Sea as a whole. The ecosystem structure and limiting factors are quite different in the North Sea and Baltic Sea: silicate limitation for diatoms in the North Sea (Reid et al., 1990), low salinity to allow cyanobacteria growth in the Baltic Sea (Wasmund, 1997). In addition, the physics model needs to performed well in both seas.

Maar et al. (2011) did a combined physical biogeochemical model simulation with HBM-ERGOM covering the North Sea and Baltic Sea. The ERGOM configuration in their study was a bit different. Their model setup performed better in some aspects – e.g. their chlorophyll-a concentrations were closer to measurements. However, they also did not fully reproduce the DIN depletion in the southern North Sea in summer in two of three years (Maar et al., 2011, Fig. 10 center left).

     To cover the North Sea and Baltic Sea ecosystem dynamics, a more complex biogeochemical model – including the sediment

– would probably be necessary. But, for this study we wanted to use a slim biogeochemical model: all nitrogen-containing tracers are duplicated for each tagged source. Therefore, the number of tracers should not be too high. The advection and diffusion of each tracer costs a lot of run time and, hence, low tracer numbers were favorable for this study.

     Other studies used coarsely resolved nitrogen deposition data like monthly or annual averages (e.g., Große et al., 2017; Troost et al., 2013), which do not capture the spatiotemporal variability of nitrogen deposition – e.g. the patchy spatial pattern

of nitrogen wet deposition. Often $50 \times 50\,\mathrm{km}^2$ resolved EMEP nitrogen deposition data are used, which is problematic in coastal areas as noted above. The only Baltic Sea study by Raudsepp et al. (2013) uses nitrogen deposition data of the Finish Hilatar CTM with $\approx 7\,\mathrm{km}$ ($0.068°$) horizontal resolution, but is limited to the Gulf of Finland. This study uses daily averaged nitrogen deposition data of $16 \times 16\,\mathrm{km}^2$ resolution, which is a considerable improvement compared to other previous studies of this large spatial coverage.

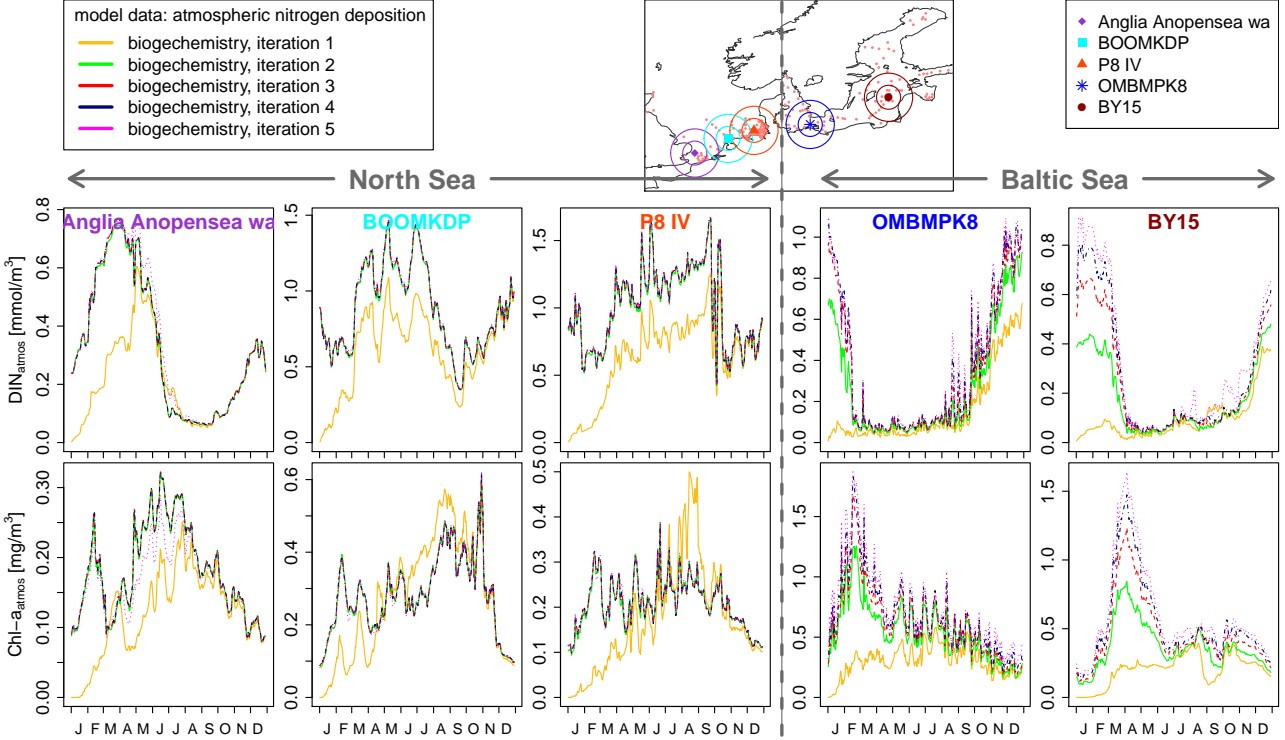

**Figure 11.** Contribution of atmospheric deposition to DIN (top) and chlorophyll-a (bottom). The atmospheric contribution in five consecutive years is plotted (see legend top left) at five locations (see legend top right). The order of the time series plots equals the order of the stations on the map from the left to the right.

Summarizing, while not all model variables are well represented in the model runs, the model covers the North Sea and Baltic Sea, and the nitrogen deposition data is considerably higher resolver than in other studies. Nevertheless, we need to be very careful not to over-interpret the results in the German Bight due to the lacking DIN depletion in summer.

## 3.3 Propagation of atmospheric contribution over five years

Five iterations of one year were simulated for evaluating the atmospheric contribution (see Sect. 2.2.4). The concentrations of tagged tracers in the end of one iteration were supplied as initial conditions for the next iteration. Simulating only one year, which starts from untagged conditions, is not sufficient because nitrogen from atmospheric sources remains in the surface layer of the Baltic Sea for more than one year and needs to reach a steady-state (see also Los et al. (2014)). Figure 11 shows concentrations of tagged nitrogen from atmospheric deposition in DIN and in chlorophyll-a at five stations over five annual iterations. Data at the same stations as in the validation section are plotted.

At P8 IV, the $DIN_{atmos}$ time series of the iteration 2 (2nd year) does only marginally deviate from the time series' of the subsequent iterations. Also the time series of iteration 1 does not differ much (1st year). In none of the iterations $DIN_{atmos}$ is



depleted in summer. In the validation section it was already noted that nitrate depletion in summer is not reproduced by the model. Hence, the prevalence of $DIN_{atmos}$ represents a model artifact.

Similar to $DIN_{atmos}$, the $Chl-a_{atmos}$ intra-annual cycles of the iteration 2 and onwards are very similar. In contrast to DIN, the $Chl-a_{atmos}$ annual cycle of iteration 1 differs considerably from the cycles of the later iterations. While $Chl-a_{atmos}$ has an

increasing trend until late August in the iteration 1, there is no such trend in later iterations. This is caused by a considerably weaker flagellate bloom in iteration 2 and onwards. Throughout the years, the relative contribution of $Chl-a_{atmos}$ to $Chl-a_{total}$ is similar during this bloom. Contrary, $Chl-a_{atmos}$ peaks in February of iteration 2 but is absent in the iteration 1. Diatoms bloom in this period in all iterations. However, no atmospheric nitrogen is incorporated into the diatoms in the iteration 2 because the $DIN_{atmos}$ concentrations are too low.

The relation between chlorophyll-a concentrations of iteration 1 is not identical to later iterations at the three North Sea stations. However, built-in nitrogen from atmospheric deposition approaches nearly a steady-state after one year at each station in the North Sea. Thus, one year of spin-up is sufficient for the North Sea. This assumption is not generally valid but only for the given model setup.

In contrast in the Baltic Sea, the winter and autumn $DIN_{atmos}$ concentrations rise iteration by iteration. They seem to

converge in iteration 5 at OMBMPK8 but they further increase at BY15. During summer, they are depleted in each year. $Chl-a_{atmos}$ concentrations rise within the first iterations but then become stagnant in the western Baltic Sea (OMBMPK8) and further rise in the central Baltic Sea (BY15).

The two considered stations OMBMPK8 and BY15 are representative for the Belt Sea, the Bay of Mecklenburg, and the Arkona Basin (OMBMPK8) as well as for the Eastern and Western Gotland Basins (BY15). Both have in common that the

modeled $DIN_{atmos}$ concentrations rise within the iterations. This increase does not only take place in the surface layer, which was plotted in Fig. 11, but also in the bottom layer (not plotted). Thus, the $DIN_{atmos}$ concentrations evolve differently than in the North Sea.

A reason for the difference between North Sea and Baltic Sea is that the water of the German Bight and dissolved nutrients are flushed on annual time scales (Lenhart and Pohlmann, 1997; Beddig et al., 1997; Pätsch et al., 2010; Pätsch et al., 2018)

either into the Northeast Atlantic along the Norwegian coast or into the Kattegat. Contrary, the residence time of water masses and dissolved nutrients in the Baltic Sea is considerably longer. In the Baltic Sea, nutrients are primarily carried out via sedimentation and not via outflow into the North Sea. The residence time of nitrogen in the water column is in the order of several years and that of phosphorus in the order of a few decades (Radtke et al., 2012). Therefore, the last iteration is better suited than earlier ones for an evaluation of the nutrient contribution by atmospheric deposition.

Figure 11 indicates that a steady-state of tagged atmospheric nitrogen in DIN and chlorophyll-a is not reached after the fifth iteration yet. Therefore, further iterations would be necessary. Originally, we simulated ten iterations. But, during the sixth and seventh iteration some tracers in deeper layers and in the eastern Baltic Sea did not behave as they should. This might be related to the simple sediment representation in the model, to problems with the vertical mixing, or to the fact that we repeated the same external forcing and physics each year. Because we did not have the capacity to further evaluate the issue





and because HBM-ERGOM is made for operational purposes and not made to run freely for a decade, we did not try to improve the long-term stability of the model but decided to consider only the first five iterations.

## 4    Conclusions

Five years of HBM-ERGOM model simulations with tagged atmospheric nitrogen deposition were performed. Not five con-
secutive years were simulated but one year was repeatedly simulated for five times – denoted as five iterations numbered from 1 to 5. The nutrient concentrations at the end of one iteration were used as initial conditions for the next iteration except for silicate, which was restored each year. Iteration 1 was used for the model validation and all five iterations for the evaluation of the atmospheric nitrogen contribution.

HBM-ERGOM reproduced measurements and the general system behavior fairly well at Baltic Sea stations at the sea surface
in the validation year. Preliminary model runs yielded an issue of steadily declining silicate concentrations over the iterations. It is not clear whether this was caused by input data – too low silicate input – or by missing recovery processes. To avoid the model running into an unrealistic silicate limitation, the silicate concentrations were restored each iteration to the conditions after the spin-up phase in the productive model runs for this study. Bottom layer oxygen minimum regions were not properly reproduced in the central Baltic Sea. This might result from a too short model run time or from overestimated vertical mixing.
The high measured ammonium concentrations in depth below $100$ m were not reproduced by the model. In the North Sea, a diatom spring bloom did not evolve properly because silicate concentrations were probably too low. As a result, DIN was not depleted in spring. Additionally, too little DIN was removed because denitrification in Wadden Sea areas was not properly represented. It is not clear in which way this affected the flagellate blooms. Despite of these shortcomings, we assume that the relative contribution of atmospheric nitrogen deposition to the marine nitrogen budget was properly reproduced. However,
these shortcomings should be evaluated in detail in future studies.

In evaluation of the atmospheric contribution, only surface water concentrations and no vertical data were considered. The model setup included a simple one-layer sediment model, which contained only two tracers (nitrogen and silicon). Hence, it is not clear how well bottom layer tracer concentrations are reproduced due to interaction with the sediment. No processes for iron reduction and phosphate release under anoxic conditions in the sediment are included (Gustafsson and Stigebrandt, 2007;
Sundby et al., 1992). We have no evidence that surface layer concentrations are impacted by these simplifications. In future studies, the simple sediment model should be replaced by multi-layer sediment including more relevant sediment tracers and processes.

The DIN and chlorophyll-a concentrations with tagged atmospheric nitrogen reached a steady-state after two iterations in the North Sea. In subsequent iterations these concentrations did not change. The North Sea is flushed each year or two (Lenhart
and Pohlmann, 1997). As a result, the residence times of nutrients in the southern North Sea are also in the order of one year and below (Beddig et al., 1997; Pätsch et al., 2010; Pätsch et al., 2018). Hence, the result that a steady-state is reached after the second iteration is reasonable.



In the Baltic Sea, the DIN and chlorophyll-a concentrations with tagged atmospheric nitrogen increased iteration by iteration. The concentrations of the fourth and fifth iteration were close to each other but a steady-state was not fully reached yet. We consider it to be sufficiently close. Other studies indicated that the residence time of nitrogen in the Baltic Sea is in the order of several years. The mean residence time of riverine nitrogen was estimated to be 1.4 years, whereas the mean residence time

nitrogen – independent of its source – in the western Baltic Sea was found to be in the order of four years (Radtke et al., 2012). Based on these results, atmospheric nitrogen should have a residence time above four years, which is consistent with this study's results. These results imply that North Sea and Baltic Sea model studies have to consider differently long time periods. Particularly tagging studies in the Baltic Sea should have a spin-up time of more than four years in order to properly capture the residence time of atmospheric nitrogen deposition. Moreover, long term simulations are necessary to capture the effects

of changes of atmospheric deposition by emission reductions – e.g. to evaluate impacts of emission reductions through legal emission thresholds. Finally, the time spans given in this study are only valid for atmospheric nitrogen deposition. Radtke et al. (2012) found that the residence time of riverine phosphorus is considerably higher than that of riverine nitrogen. Corresponding to the results of Radtke et al. (2012), atmospheric phosphorus probably also has longer residence times than atmospheric nitrogen.

Finally, based on the results of this first part of our study it is recommended to evaluate the atmospheric nitrogen contribution in more detail using simulations of five and more repeated years and compare the results with other published studies. Therefore, results of iteration five are the basis for the evaluation in the second part of this study (Neumann et al., 2018b). Moreover, the tagging approach offers the possibility to trace the deposited nitrogen of specific anthropogenic emission source sectors – namely the shipping and the agricultural source sectors – in the second part of this study.

*Code and data availability.* .

**Model Code:** The original HBM-ERGOM code was provided by the Federal Maritime and Hydrographic Agency of Germany (BSH). The license agreement does not allow the authors to pass the code to third parties. The code can be requested from the BSH or the Danish Meteorological Institute (DMI). The modified ERGOM code and brief description of the model processes and constants are attached in the supplement.

**Model output data:** The data are available via the THREDDS server of the IOW: https://thredds-iow.io-warnemuende.de/thredds/projects/meramo/catalog_meramo_cmaq16_silrestart.html

**Measurement data:**

  – HELCOM data are available via the ICES homepage: http://ocean.ices.dk/helcom/Helcom.aspx

  – IOWDB data are available on request (https://www.io-warnemuende.de/iowdb.html). Please contact to authors to get access to the
database.

  – ICES data are available via the ICES Oceanography data database http://ocean.ices.dk/HydChem/HydChem.aspx

  – DOD data are available on request via the DOD Cruise Data Mining portal http://seadata.bsh.de/csr/retrieve/dod_index.html

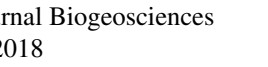



*Author contributions.* .

**Daniel Neumann:** overall structure; HBM-ERGOM model simulations; programming work; plotting; major writing tasks

**Matthias Karl:** CMAQ air quality model simulations; evaluation of meteorological forcing data and of nitrogen deposition data; contribution to Materials & Methods and Results & Discussion sections; development of research question

**Hagen Radtke:** implementation of the tagging method; framework for data processing; contribution to Results & Discussion and Materials & Methods sections; discussions during data evaluation; development of research question

**Thomas Neumann:** development of the research question; contribution to Introduction, Materials & Methods and Conclusions

*Competing interests.* The authors declare that they have no conflict of interest.

*Acknowledgements.* Parts of the research published in this publication were carried out in the research projects MeRamo (funded by BMVI,
FKZ 50EW1601) and BONUS SHEBA (Sustainable Shipping and Environment of the Baltic Sea region). The BONUS SHEBA project was
supported by BONUS (Art 185), funded jointly by the EU and national funding institutions. The HBM-ERGOM model simulations were
performed at the cluster Gottfried of the North-German Supercomputing Alliance (HLRN, project ID mvk00054). The meteorological and
atmospheric chemistry transport model (CTM) simulations were performed at the German Climate Computing Center (DKRZ) within the
Project "Regionale Atmosphärenmodellierung" (Project ID 0302), which is funded by the Helmholtz Association. The emissions for the
air quality model simulations were kindly provided by Johannes Bieser, Armin Aulinger, and Jukka-Pekka Jalkanen. The HBM is currently
maintained by the Danish Meteorological Institute (DMI) and the Federal Maritime and Hydrographic Agency of Germany (BSH), namely
Thorger Brünning. The air quality model CMAQ is developed and maintained by the U.S. Environmental Protection Agency (US EPA).
We thank our colleagues conducting IOW's Baltic Monitoring and long-term data program, which intense quality checked measurements
we used for the model validation. Some of the measurement data were kindly provided by the HELCOM oceanographic measurements
database hosted by ICES. Martin Schmidt of the IOW supported us with respect to preparation and upload of model data to the IOW
THREDDS server. Anja Eggert commented the plotting and provided information on phytoplankton growth. Johannes Pätsch provided value
input on the residence time of nutrients in the German Bight. We thank Uwe Schulzweida, Charlie Zender, Paul Wessel, the R Core Team,
and the Unidata development team (and all involved developers/contributors) for maintaining the open source software packages Climate
Data Operators (cdo), the NetCDF Operators (NCO), Generic Mapping Tools (GMT), the statistical computing language R, and netCDF,
respectively.





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
