# Peer review of "Evaluation of atmospheric nitrogen inputs into marine ecosystems of the North Sea and Baltic Sea – part A: validation and time scales of nutrient accumulation"

_Biogeosciences, 2018_

## Referee Comment (RC1) · Anonymous Referee #2 · 1 Nov 2018

The study aims to validate a marine ecosystem model and to quantify times scales of nutrient accumulation (c.f. the title of the study). The times scales of nutrient accumulation are expressed in terms of a residence time. The focus of the study is on the residence times of atmospheric bioavailable nitrogen in the North Sea and the Baltic Sea. Model results presented suggest that the respective residence times are two years in the North Sea and five years in the Baltic. These results are consistent with already-published estimates (c.f. pg. 1, ln. 16-18).

My main criticism is that, to calculate a residence time, a model is not needed. Multipli-

cation of observed nutrient inventories with the inverse of the HELCOM nutrient fluxes directly, at the back of an envelope, yields residence times already. According to the authors (c.f. pg. 1, ln. 16-18) these residence times have been already known. I conclude that their model estimate does not present novel concepts, ideas - nor substantial conclusions.

The authors state that simulated deep nutrient concentrations in the Baltic are biased and that denitrification in the Wadden Sea is underestimated but that, at the same time, that " ... this did not impact surface layer concentrations" (pg. 1, ln. 13 to 15). Assuming that simulated surface nutrient concentrations were realistic makes me wonder if they are so for realistic reasons.

As concerns the second aim of the study, to validate a marine ecosystem model, I feel that a more specialized journal like "Geophysical Model Development" would be more appropriate because the audience addressed by Biogeosciences is rather broad.

I recommend to reject the manuscript and encourage resubmission to a journal that is focused on model description and validation.

---

## Author Comment (AC1) · 9 Nov 2018

We thank the anonymous referee 2 for the review of the discussion paper "Evaluation of atmospheric nitrogen inputs into marine ecosystems of the North Sea and Baltic Sea – part A: validation and time scales of nutrient accumulation".

We would like to directly reply to the second concern of referee 2 reading:

> *As concerns the second aim of the study, to validate a marine ecosystem model, I feel that a more specialized journal like "Geophysical Model Development" would be*

[Figure]

*more appropriate because the audience addressed by Biogeosciences is rather broad.
I recommend to reject the manuscript and encourage resubmission to a journal that is
focused on model description and validation.*

We agree with referee 2 that this study ("part A") is clearly focused on the model evaluation/validation. When considered individually, it is more appropriate for journals such as GMD. However, we submitted it in combination with a second discussion paper ("part B"), which is focused on the contribution of nitrogen deposition from different atmospheric emission sources to surface DIN and PON concentrations. Unfortunately, part B of the study (doi: https://dx.doi.org/10.5194/bg-2018-365) was not available online when the review of referee 2 was performed.

Originally, both discussion papers were one manuscript, which was very long. Therefore, we decided to split it into two short – still long – discussion papers consisting of part A (model validation and first results) and part B (detailed results and evaluation). We have the feeling that both discussion papers belong together. Discussion paper part B without validation of the model would be questionable. Hence, we hesitate(d) to submit them to two different journals.

We are the users but not the developers of the particular model version, which was used for this study. The actual developers should be the ones to published a detailed validation of their model – getting the credits (and a first-author publication) for the development work. Hence, we limited the validation to the year and to the aspects, which are relevant for our evaluation of nitrogen deposition data, leaving the developers the possibility a publish a full validation in an appropriate journal. If such a publication would be available, we would have omitted submitting discussion manuscript part A and would only have submitted part B (without "part B" in the title).

---

## Referee Comment (RC2) · Anonymous Referee #3 · 3 Dec 2018

This manuscript presents and evaluates a coupled physical biogeochemical model HBM-ERGOM forced by modelled atmospheric deposition of nitrogen for its ability to simulate dissolved inorganic nitrogen (DIN), dissolved inorganic phosphorus, silicate, oxygen, and chlorophyll-a in the seawater of the North and the Baltic Seas. With the aim to be used (in a companion paper) to evaluate the impact of the deposition flux from shipping and agricultural emissions of N to the marine ecosystems, the model is tagging the nutrients in the seawater and their penetration into the ecosystem components, i.e. uses source specific nutrients to evaluate their propagation in the marine

environment and their impact. This approach is very interesting and the information that could be derived from its proper implementation is expected to increase our understanding of the environmental impacts of anthropogenic N inputs to the ocean.

The main result presented in this manuscript is the residence time of N and P in the studied region, which agrees with the existing literature. This somehow provides some indications that the model is not totally unrealistic.

However, the model as presented has a number of important shortcomings also mentioned by reviewer #2, which show significant deficiencies in the model's ability to simulate the marine N cycle. Lacking DIN depletion in summer (p. 20, line 3), a period over which it is expected that the atmospheric deposition will maximize because of the stratification of the seawater, will definitely introduce large inaccuracies in the calculated impact on the marine ecosystem. Therefore, as presented the quality of the modeling is questionable.

However, the authors use the first year of their simulation to validate their model, while they clearly say in the manuscript that their model did not reach a steady-state and they finally use the 2nd or 5th year to further investigate the impacts of atmospheric deposition. I would expect to see a model validation for the iteration that is used for the impact study, since the others appear as spin-up time for their model system. This might provide totally different results for the model evaluation.

In case that the last iteration is better representing the N marine cycle than what is actually shown in the manuscript and discussed, then it might be worth considering publishing this work to BG if it fulfills my comments below or to another journal, more appropriate for model description.

The manuscript has to be significantly shortened, focused and needs clarification in several parts. The key message of the manuscript has to be the evaluation and quantification of the uncertainty in the calculations and has to be reorganized in this direction. Several parts of the present manuscript can move to the supplement. To be suitable

for BG the manuscript needs also to further elaborate the science question, i.e. the contribution of atmospheric deposition to the DIN in the seawater (based on figure 11 and Figures 6&9). The authors might consider merging it with the companion paper.

Further comments for potential improvements: From all the figures here presented, figures 5, 6, 9 and 11 are the most informative for the purpose of the surface validation discussed here. In figure 6 and 9, I think the simulation that will be used for the impact study has to be evaluated and not the first one. Fig 11 shows the tagged DIN but then it is not a contribution, for contribution one is expecting to see a ratio or a % value to the total DIN in the seawater. Also consider merging parts of Fig 6/9 and 11? Table 2 and 3 could be merged and additional literature data could be added for comparison. Finally, It is often confusing whether DIN concentrations in the atmospheric deposition or in the seawater are discussed/ shown in the figures. Also figures caption have to be more informative, e.g. is surface seawater composition shown or something else?

---

## Short Comment (SC1) · 11 Dec 2018

Reviewer's comments to manuscripts by Neumann, D., Karl, M., Radtke, H., and Neumann, T. "Evaluation of atmospheric nitrogen inputs into marine ecosystems of the North Sea and Baltic Sea – part A: validation and time scales of nutrient accumulation; part B: contribution by shipping and agricultural emissions" submitted to "Biogeosciences"

The study aims at a detailed quantitative description of the pathways and effects of

atmospheric nitrogen inputs in the marine ecosystems of the North and Baltic seas as simulated with the coupled physical-biogeochemical model HBM-ERGOM. The sine qua non precondition for achieving such ambitious, if somewhat artificial, goal is the realistic simulation of biogeochemical nitrogen cycling in both marine systems. That's why both manuscripts must be considered together, starting from the model itself. Unfortunately, the implemented model version is not suitable for such studies in many aspects: A) by deficient formulations; B) by failing in reproducing some phenomena crucially important in nitrogen cycling; C) by flawed set-up of numerical experiments and validation; and, finally, D) by poor model-data comparability. All these, taken together convert presented results in merely casual exercises that have little to do with the realistic cycling of atmospheric nitrogen in marine ecosystems. That's why I would not even go further into detailed reviewing of "tagged" results. Instead, I recommend to reject both manuscripts and advice against using this version of HBM-ERGOM model, made for operational purposes (perhaps, with the data assimilation), for the long-term studies.

A few examples of crucial flaws and drawbacks are given below.

A) "Iron reduction and release of phosphate under anoxic conditions in the sediment are not represented in this ERGOM version" (Part A, L 15/8). Fixing sediment N:P ratio and ignoring redox alterations of the P cycle implausibly affects phosphate dynamics, hence, distorts such important flux as nitrogen fixation and the following cycling of fixed nitrogen. The necessity of Si restarting for every year indicates that its dynamics even during the first iteration is erroneous with corresponding consequences for phytoplankton seasonal succession and nutrient uptake. Finally, many important features and phenomena, for instance, nutrient limitation, nutrient residence times, species composition, tides and oceanic impacts, etc., are rather different between the North and Baltic seas. That makes combining them into a single domain questionable, if not harmful for the objectives of this study.

B) Overestimated deep layers oxygen concentration and underestimated denitrification

distort DIN distribution and dynamics (see comparisons in Figs. 7-10). Together with questionably reproduced nitrogen fixation, such underestimation indicates a wrong balance between nitrogen sources and sinks, hence, biases evaluation of atmospheric N contribution to unknown degree.

C) "Therefore, a detailed validation of the nitrogen deposition data sets is not possible and it is not clear whether the CMAQ nitrogen deposition is actually too low over sea." (Part A, L16-18/13). Already this statement makes studies of the RELATIVE contributions rather uncertain. Further uncertainty (due to possible non-linear effects in the biogeochemical cycling) is introduced by the repetitive implementation of deposition computed only for one year (i.e. 2012) over all five years, forcing a possible deficit accumulation.

D) The model set-up and simulated dynamics contain many features that are "typical within order of magnitude" rather than year-specific. Therefore a comparison of the "first" iteration with observations during concrete 2012 year looks very optimistic, even naïve. Perhaps, such choice partly explains why most patterns of seasonal dynamics are very poorly reproduced either in timing or by the levels, or both (Figs. 7-10). Never mind the plausible oxygen dynamics in the surface layer, where it is mainly driven by air-sea gas exchange. Moreover, the focusing of analysis at the surface layer is unwarranted because the nitrogen biogeochemical cycle must be evaluated for the entire ecosystem, including sediments.

---

## Author Comment (AC2) · 12 Dec 2018

**Response to review comment #1 by referee #2**

We thank the reviewer for the constructive comments on the manuscript.

Below, the reviewer's comments are written in bold letters and our answers in non-bold letters.

[Figure]

**My main criticism is that, to calculate a residence time, a model is not needed. Multiplication of observed nutrient inventories with the inverse of the HELCOM nutrient fluxes directly, at the back of an envelope, yields residence times already. According to the authors (c.f. pg. 1, ln. 16-18) these residence times have been already known. I conclude that their model estimate does not present novel concepts, ideas – nor substantial conclusions.**

> We agree with the reviewer that one can estimate the residence times by back-of-the-envelope calculations. However, the residence times of nutrients of individual sources depend on the spatio-temporal input pattern. Nitrogen compounds sourced in flat coastal regions are removed faster by denitrification than nutrients sourced in deep open basins. Inorganic nutrients sourced during summer are faster processed by phytoplankton than nutrients sourced during early winter. Hence, we think that it is important to include the spatio-temporal variability of nitrogen inputs in the estimation of nitrogen residence times.

> This manuscript rather is to see as a basis for the companion paper part B.

**The authors state that simulated deep nutrient concentrations in the Baltic are biased and that denitrification in the Wadden Sea is underestimated but that, at the same time, that " ... this did not impact surface layer concentrations" (pg. 1, ln. 13 to 15). Assuming that simulated surface nutrient concentrations were realistic makes me wonder if they are so for realistic reasons.**

> Because the model performance was low in the German Bight, we plan to completely remove the North Sea from the manuscript. The evaluation of the model results of the Baltic Sea also revealed issues but these are not as severe as in the North Sea.

**As concerns the second aim of the study, to validate a marine ecosystem model, I feel that a more specialized journal like "Geophysical Model Development"**

**would be more appropriate because the audience addressed by Biogeosciences is rather broad.**

> We agree with the referee that this study ("companion paper part A") is clearly focused on the model evaluation/validation. When considered individually, it is more appropriate for journals such as GMD. However, we submitted it in combination with a second discussion paper ("part B"), which is focused on the contribution of nitrogen deposition from different atmospheric emission sources to surface DIN and PON concentrations. Unfortunately, part B of the study (doi: https://dx.doi.org/10.5194/bg-2018-365) was not available online when the review of the referee was performed.

> Originally, both discussion papers were one manuscript, which was very long. Therefore, we decided to split it into two shorter – still long – discussion papers consisting of a *part A* (model validation and first results) and *part B* (detailed results and evaluation). We have the feeling that both discussion papers belong together. Discussion paper part B without validation of the model would be questionable. Hence, we hesitate(d) to submit them to two different journals.

> We are the users but not the developers of the particular model version, which was used for this study. The actual developers should be the ones to published a detailed validation of their model – getting the credits (and a first-author publication) for the development work. Hence, we limited the validation to the year and to the aspects, which are relevant for our evaluation of nitrogen deposition data, leaving the developers the possibility a publish a full validation in an appropriate journal. If such a publication would be available, we would have omitted submitting discussion manuscript part A and would only have submitted part B (without "part B" in the title).

---

## Author Comment (AC3) · 12 Dec 2018

**Response to review comment #2 by referee #3**

We thank the reviewer for the constructive comments on the manuscript.

Below, the reviewer's comments are written in bold letters and our answers in non-bold letters.

[Figure]

The main result presented in this manuscript is the residence time of N and P in the studied region, which agrees with the existing literature. This somehow provides some indications that the model is not totally unrealistic.

However, the model as presented has a number of important shortcomings also mentioned by reviewer #2, which show significant deficiencies in the model's ability to simulate the marine N cycle. Lacking DIN depletion in summer (p. 20, line 3), a period over which it is expected that the atmospheric deposition will maximize because of the stratification of the seawater, will definitely introduce large inaccuracies in the calculated impact on the marine ecosystem. Therefore, as presented the quality of the modeling is questionable.

> We plan to remove the North Sea from the manuscript and consider only the Baltic Sea because the severe issues occur in the North Sea.

However, the authors use the first year of their simulation to validate their model, while they clearly say in the manuscript that their model did not reach a steady-state and they finally use the 2nd or 5th year to further investigate the impacts of atmospheric deposition. I would expect to see a model validation for the iteration that is used for the impact study, since the others appear as spin-up time for their model system. This might provide totally different results for the model evaluation.

In case that the last iteration is better representing the N marine cycle than what is actually shown in the manuscript and discussed, then it might be worth considering publishing this work to BG if it fulfills my comments below or to another journal, more appropriate for model description.
> We thank the reviewer for the suggestion to consider later years for the validation. Unfortunately, the situation does not improve in the North Sea in later years but seems to be quite stable from the 2nd year and onwards.

**The manuscript has to be significantly shortened, focused and needs clarification in several parts. The key message of the manuscript has to be the evaluation and quantification of the uncertainty in the calculations and has to be reorganized in this direction. Several parts of the present manuscript can move to the supplement.**

> We will consider these comments in the revision.

**To be suitable for BG the manuscript needs also to further elaborate the science question, i.e. the contribution of atmospheric deposition to the DIN in the seawater (based on figure 11 and Figures 6&9).**

> We will work on the science question.

**The authors might consider merging it with the companion paper.**

> We will consider merging both manuscripts if this is possible in this publication format.

**Further comments for potential improvements: From all the figures here presented, figures 5, 6, 9 and 11 are the most informative for the purpose of the surface validation discussed here. In figure 6 and 9, I think the simulation that will be used for the impact study has to be evaluated and not the first one. Fig**

**11 shows the tagged DIN but then it is not a contribution, for contribution one is expecting to see a ratio or a % value to the total DIN in the seawater. Also consider merging parts of Fig 6/9 and 11? Table 2 and 3 could be merged and additional literature data could be added for comparison. Finally, It is often confusing whether DIN concentrations in the atmospheric deposition or in the seawater are discussed/ shown in the figures. Also figures caption have to be more informative, e.g. is surface seawater composition shown or something else?**

> We will consider these comments in the revision.

———————————————

---

## Author Comment (AC4) · 12 Dec 2018

**Response to short comment #1 by Oleg Savchuk**

We thank Oleg Savchuk very much for reading this and the companion manuscript. We agree with most of the four major critical aspects mentioned and cannot satisfy/disprove them completely.

We posted a reply to his comment in the discussion of the companion paper part B:

[Figure]

- discussion paper part B: https://doi.org/10.5194/bg-2018-365

- direct URL to our reply: https://editor.copernicus.org/index.php/bg-2018-365-AC3.pdf?_mdl=msover_md&_jrl=11&_lcm=oc108lcm109w&_acm=get_comm_file&_ms=70639&c=153438&salt=8290753261258637551

---

## Editor Comment (EC1) · Sarin (Editor) · 9 Jan 2019

This study aims to evaluate the contribution of atmospheric nitrogen deposition to marine ecosystems of North Sea and Baltic Sea emphasizing on the residence time of N. Authors state that "the concentrations of dissolved and particulate nitrogen in the sea are not only determined by the input, but also by the residence time of nitrogen in the system before it is removed by biogeochemical processes or physical advection". This is miss-leading. The concept of residence time is based on concentration of nutrient in the steady-state and rate of input or removal from the ecosystem. Thus, use of model

to validate time-scales of nutrient accumulation is not well understood. Moreover, Authors have arrived at some obvious conclusions that results are consistent with the published residence time of nutrients. Nevertheless, the use of model has some basic limitations with respect to spatio-temporal variability in simulating DIN and impact on marine ecosystem. The simulation and validation of the model to attain steady-state is not adequately explained with respect to atmospheric deposition of nitrogen. The evaluation of uncertainties associated with the model needs better approach and quantification. Overall, evaluation of biogeochemical cycling of nitrogen (and nutrients) is not rigourously built in the manuscript with respect to sources other than atmospheric deposition (example, redox conditions in the sediments). Relying only on surface water concentrations does not meet the objectives. Authors have discussed number of shortcomings, but still assume that "the relative contribution of atmospheric nitrogen deposition to the marine nitrogen budget was properly reproduced". Authors also believe these shortcomings should be evaluated in detail in future studies. The reference made to second part of study further raises limitation on the suitability of the existing model

---

## Author Comment (AC5) · 9 Jan 2019

We thank the Editor Manmohan Sarin for taking time to moderate the review process and for providing a final comment on the manuscript. We agree with the criticism and consider to replicate the study with a better suited biogeochemical model over longer time scales (and re-submit it as a new publication).